# Kidney function in healthcare clients in Khayelitsha, South Africa: Routine laboratory testing and results reflect distinct healthcare experiences by age for healthcare clients with and without HIV

**Richard Osei-Yeboah**[1], **Olina Ngwenya**[2,3], **Nicki Tiffin**[3,4]*

1 Faculty of Health Sciences, Integrative Biomedical Sciences Department, Division of Computational Biology, University of Cape Town, Cape Town, South Africa, 2 Faculty of Biology, Centre for Biostatistics, School of Health Sciences, Medicine and Health, University of Manchester, Manchester, United Kingdom, 3 Institute of Infectious Diseases and Molecular Medicine, Wellcome Centre for Infectious Disease Research in Africa, University of Cape Town, Cape Town, South Africa, 4 South African National Bioinformatics Institute, South African Medical Research Council Bioinformatics Unit, University of the Western Cape, Bellville, South Africa

* ntiffin@sanbi.ac.za

**Data Availability Statement:** These anonymised, perturbed data were provided for analysis by the

## Abstract

In South Africa, PLHIV are eligible for free ART and kidney function screening. Serum creatinine (SCr) laboratory test data from the National Health Laboratory Service are collated at the Provincial Health Data Centre and linked with other routine health data. We analysed SCr and estimated glomerular filtration rate (eGFR) results for PLHIV and HIV-negative healthcare clients aged 18–80 years accessing healthcare in Khayelitsha, South Africa and comorbidity profiles at SCr and eGFR testing. 45 640 individuals aged 18–80 years with at least one renal test accessed Khayelitsha public health facilities in 2016/2017. 22 961 (50.3%) were PLHIV. Median age at first SCr and eGFR test for PLHIV was 33yrs (IQR: 27,41) to 36yrs (IQR: 30,43) compared to 49yrs (IQR: 37,57) and 52yrs (IQR: 44,59) for those without HIV. PLHIV first median SCr results were 66 (IQR: 55,78) μmol/l compared to 69 (IQR: 58,82) μmol/l for HIV-negative individuals. Hypertension, diabetes, and CKD at testing were more common in HIV-negative people than PLHIV. HIV, diabetes and tuberculosis (TB) are associated with higher eGFR results; whilst hypertension, being male and older are associated with lower eGFR results. These data reflect testing practices in the Western Cape: younger people without HIV have generally worse kidney function test results; younger PLHIV have generally good test results, and older people with/without HIV have generally similar test results, reflecting regular screening for kidney function in asymptomatic PLHIV whereas young HIV-negative people are tested only when presenting with renal symptoms. Our analysis suggests we cannot infer the future healthcare requirements of younger PLHIV based on the current ageing population, due to changing ART availability for different generations of PLHIV. Instead, routine health data may be used in an agile way to assess ongoing

Western Cape Department of Health, Provincial Health Data Centre. These are highly granular health data linked to individual health care clients in the province and no informed consent has been given for research use. For this reason, the Western Cape Department of Health does not permit open sharing but instead grants only primary use permission for the data. Re-use of this dataset requires approval from the Western Cape Department of Health (Provincial Health Data Centre), and Dr Moodley, Director: HIA, Western Cape Department of Health, South Africa can be contacted to advise on this process (email: melvin.moodley@westerncape.gov.za, Reference study ID 259-TIFFIN).

**Funding:** This research was funded in part, by the Wellcome Trust 203135/Z/16/Z. For the purpose of open access, the author has applied a CC BY public copyright licence to any Author Accepted Manuscript version arising from this submission. The Wellcome Centre for Infectious Diseases Research in Africa is supported by core funding from the Wellcome Trust [203135/Z/16/Z]. ROY receives funding from the Africa Regional Staff/ Student International Exchange: Food Security and Sustainable Human Wellbeing II (ARISE II) Consortium funded by the Intra-Africa Academic Mobility Scheme of the European Union (2016-2616). NT received funding from Wellcome CIDRI-Africa (203135/Z/16/Z) and ON was supported by this grant. NT receives funding from UKRI/MRC (MC_PC_22007) and is a recipient of a Calestous Juma fellowship from the Bill & Melinda Gates Foundation (INV-037558). The funders had no role in study design, data collection and analysis, decision to publish, or preparation of the manuscript.

**Competing interests:** The authors have declared that no competing interests exist.

healthcare requirements of ageing PLHIV, and to reflect implementation of treatment guidelines.

## Introduction

People living with Human Immunodeficiency Virus (PLHIV) may have a wide range of kidney conditions which could be as a result of the HIV infection itself, long-term exposure to antiretroviral therapy (ART), or treatments and effects from other chronic comorbidities which also affect HIV-negative individuals [1]. HIV-associated nephropathy (HIVAN) is one of the most common renal disorders among PLHIV [1,2] and several factors including comorbidities, high viral load, low CD4+ counts and advanced kidney disease have been identified as risk factors for the occurrence of HIVAN and subsequent progression to end-stage renal disease (ESRD) [3,4]. Suggested mechanisms of HIV affecting renal cells include a direct infection of the renal parenchymal cells by the virus, renal cellular uptake of circulating virally encoded molecules causing indirect injury to the kidney, or indirect injury from the release of cytokines [3]. Although the prevalence of HIVAN has declined due to improved ART treatment modalities and coverage, studies suggest that the risk of other renal disorders, including chronic kidney disease (CKD), among PLHIV is about 4 times higher compared to the general population [5,6].

The South African National Department of Health recommends a first-line HIV treatment regimen consisting of a fixed dose combination of Tenofovir disoproxil fumarate (TDF), Lamivudine (3TC) and Dolutegravir (DTG)-TLD for all eligible adults, adolescents and children over 10 years and weighing 35kg and above, and tenofovir disoproxil fumarate-emtricitabine-efavirenz (TEE) for women of childbearing potential wanting to conceive due to safety issues of TLD in the first 6 weeks of pregnancy [7]. A major issue concerning many of these agents, especially TDF, has been nephrotoxicity and these agents are associated with the development of acute and chronic kidney diseases [1,3,8,9]. Although Africa ranks highest with an estimated prevalence of CKD among PLHIV globally, at 7.9%, Southern Africa has the lowest regional prevalence estimated at 3.2% [10] and studies conducted in South Africa have reported low prevalence of renal impairment and better glomerular filtration rates among PLHIV compared to other African countries [11,12].

Diabetes mellitus and hypertension are reported major causes of kidney disease in the general population and together account for about 70% of all ESRD [3]. Considering the high prevalence of these chronic conditions in South Africa and associated long-term treatment, and the known relationship between CKD and HIV, and evidence of increased blood pressure and risk of hypertension [13] and diabetes mellitus among PLHIV on ART, both PLHIV and seronegative populations may experience the effects of these comorbidities on kidney function.

Serum creatinine (SCr) is considered the most used endogenous marker for determining kidney function and it is widely used in clinical practice [14], but its sensitivity to detect early renal disorders is questioned as it might sometimes remain at normal levels despite significant kidney impairment [15]. This may be due to the fact that the production of SCr depends on lean body mass and may not accurately reflect glomerular filtration rates, especially in older patients and females who have lower muscle mass [16,17]. It is therefore recommended that prediction equations are used to estimate glomerular filtration rate (eGFR) for reporting in conjunction with SCr to enable early detection of kidney dysfunction [14,18]. SCr may be influenced by several factors such as muscle mass, age, sex, nutritional status and chronic illnesses, so these metrics are combined in order to calculate eGFR. Measurements of SCr and

eGFR are standard laboratory results generated during routine care when monitoring patients with kidney disease, and these results can also provide a useful profile of a patient's changing renal health over time for epidemiological research.

Given the high burden of comorbidities in South African patients [19], it is important to understand the relationship of kidney disease with HIV infection in the context of other comorbidities, in order to improve patient outcomes. This study explores this relationship using comorbidity records and laboratory results for SCr and eGFR from collated routine health data to assess the association of HIV, TB, hypertension, and diabetes with kidney function in patients in the Western Cape. We have used data from the Provincial Health Data Centre (PHDC), a health information exchange that collates administrative health data from public healthcare clients in the Western Cape Province in South Africa; and the granular longitudinal laboratory test and comorbidity data provide an opportunity to link laboratory measures of kidney function with HIV and other comorbidities in these healthcare clients [20]. We describe the characteristics of healthcare clients at their first kidney function test, as well as the characteristics across all tests taken by these individuals; and explore the association between test results, demographics, and comorbidities in this population.

## Materials and methods

### Ethics

Ethics approval was obtained from the Human Research Ethics Committee of the Faculty of Health Sciences, University of Cape Town. Ethics approval for the overarching study and use of the dataset was granted on 28 May 2018 (HREC ref:282/2018). An additional ethics approval was obtained for the specific study using this dataset on 16 July 2019 (HREC ref: 482/2019). Ethics approval renewals were obtained annually until the close of the study. A data access request was additionally approved by the Health Impact Assessment Directorate at the Western Cape Department of Health, South Africa. A waiver for informed consent was granted because the data provided were anonymised and perturbed, and individuals could not be identified or re-identified from the data.

### Study population

Khayelitsha is a high-density, mixed informal and formal housing suburb in Cape Town, South Africa. The analysis includes all SCr and eGFR laboratory results generated by the National Health Laboratory Services for adults (18 years and above) who accessed public health facilities in the Khayelitsha subdistrict of Cape Town, South Africa, between 1 January 2016 and 31 December 2017, described as the 'recruitment period'.

### Data source

The PHDC is a health information exchange facility that collates administrative health data for the Western Cape Province. Unique identifiers are used to link individuals to administrative health records, and facility visit, laboratory, and pharmacy data are updated daily for about 6.6 million people currently seeking care in public facilities in the Western Cape Province. Algorithms are used to infer disease episodes from combinations of pharmacy-dispensed drugs, laboratory test results, international classification of diseases 10[th] edition (ICD-10) diagnosis codes, and facility encounter data [20]. In this study we refer to "disease ascertainment" to mean inference of the start of a disease episode as identified by an algorithm, to distinguish this process of disease ascertainment from a clinical diagnosis made by a healthcare professional during consultation. A dataset containing routine health data was obtained from the

PHDC, Western Cape Government Health Department, with longitudinal data ranging from 2007 to 2017. The dataset was compiled on 30 May 2019. The records of the healthcare clients who received laboratory results included HIV and comorbidities records that were used to determine which health conditions had already been ascertained when each kidney function laboratory test was taken. The median length of time for which individuals have available data is 8 years (Interquartile range [IQR]: 3.6–10 years). The study dataset was anonymised and perturbed within the PHDC prior to being released for use in this analysis, to prevent identification or re-identification of individuals.

## PHDC disease episode definition

The PHDC infers diseases from routine health data using algorithms that analyse either single, or a combination of parameter(s) categorised into high, moderate, weak confidence and supporting-only evidence for having a particular disease episode. High confidence definition of HIV requires evidence for dispensed valid first line (2NRTI and NNRTI) and valid triple therapy regimen (fixed-dose combination) of antiretrovirals, and/or positive laboratory test results (viral load test, polymerase chain reaction (PCR) test, enzyme linked immunosorbent assay (ELISA) test, and ART resistance test). For mycobacterium tuberculosis (MTb), evidence for the episode may include admission to a specialised tuberculosis (TB) hospital, TB drug regimen dispensed, and/or laboratory test results (Positive GeneXpert, Line probe assay (LPA), Acid-Fast Bacillus positive culture, positive microscopy (Ziehl-Neelsen staining), and microbiology culture (using ELISA) are the main definition parameters. High confidence definition of hypertension includes dispensed hydrochlorothiazide. High confidence definitions of diabetes episodes are based on dispensed drug for the treatment of diabetes mellitus, laboratory test showing glycated haemoglobin (HbA1c) greater than 6.5%, oral glucose tolerance test result greater than 11.1mmol/l, and diagnosis coding showing an ICD-10 code indicating diabetes disease.

For CKD, laboratory tests showing consecutive glomerular filtration rate of less than 60mL/min/1.73m$^2$ with 90 days between tests, dispensed kidney, or transplant medications (antithymocyte, immunoglobulin, and basiliximab), and diagnosis coding indicating kidney transplant procedure in theatre constitute a high confidence definition. It is important to note that SCr and eGFR results are used extensively in defining patients with CKD by the PHDC. This means that there is substantial overlap between having poor kidney function test results and being defined as having a CKD episode. The ascertainment algorithm in use by the PHDC, however, makes use of longitudinal eGFR results to track changes in kidney function over time, and is not based on single or stand-alone kidney function results. The PHDC algorithm uses the modification of diet in renal disease (MDRD) GFR estimating equation which adjusts for race/ethnicity to determine eGFR. We recognise that an update of the Chronic Kidney Disease Epidemiology Collaboration (CKD-EPI) equation without the race/ethnicity factor is now recommended [21,22].

## Recommendation for kidney function assessment for PLHIV in ART guideline

The South African 2019 ART Clinical Guidelines (updated 2020) recommend routine assessment of baseline kidney function before ART initiation. The guidelines recommend the use of eGFR calculated using Counahan Barratt formular for patients $\geq 10$ and $< 16$ years of age, eGFR using MDRD equation for adults and adolescents above 16 years, and absolute creatinine levels for pregnant women living with HIV before initiating ART. However, results availability prior to ART initiation is not a requirement [7].

## Data analysis

In order to understand when healthcare clients are first referred for SCr testing, we analysed the age at which first SCr and eGFR laboratory results are generated for adults (≥18 years) in the study population. The distribution of age at their first recorded SCr and/or eGFR test results according to HIV status was determined for all the healthcare clients in the study population.

Descriptive statistics were generated to summarise the counts and proportions for TB, hypertension, diabetes and CKD in healthcare clients who have SCr and eGFR testing done, by HIV status. The counts and proportions of the healthcare clients receiving eGFR results, and the ascertained comorbidities per 18–29 years, 30–44 years, 45–60 years, and over 60 years age groups were described.

SCr and eGFR are highly related parameters derived from the same tests. The National Health Laboratory Services of South Africa integrates SCr measures, age, sex, and ethnicity in estimating glomerular filtration rates [23,24]. Here, we first visualise the distribution of ages at both SCr and eGFR testing for our study population, and subsequently present analyses focusing only on eGFR to assess renal disorders in our study population, because of the inbuilt adjustment for age and sex that this measurement provides. A scatter plot distribution of all SCr and eGFR results for individuals receiving test results was generated to provide a general overview of SCr and eGFR results with respect to HIV status, age and sex for all tests taken–recognising that multiple tests may arise from a single patient in this overview. Healthcare clients without identified sex status were removed from subsequent analysis.

In order to further understand the age differences in eGFR between PLHIV and HIV-negative populations, age categories were created and a boxplot distribution of all eGFR results among the age groups at eGFR testing was generated. This provides a general overview of the eGFR results per age groups among PLHIV and HIV-negative healthcare clients receiving results.

Comparisons of eGFR results for healthcare clients with described comorbidities TB, hypertension and diabetes, were made for each age category. These comparisons were based on tests conducted on specified dates for individuals who had already been ascertained with the specified comorbidity at the time the test was done. The significance of differences between the median eGFR results was calculated using Wilcoxon rank-sum tests to generate the reported p-values.

We used a linear mixed effects model to explore the relationship between eGFR results and age at testing (by age category), sex, HIV and selected comorbidities as fixed effects. The individual-level repeated eGFR tests were included as the random effect. This method takes into account the heterogeneity which may exist in the number of kidney function estimates available for each patient which must be considered when estimating kidney function trajectories [25,26]. The model was run using the 'lme4' package in RStudio and the associated p-values were generated with the 'lmerTest' package.

Data analyses were done in R Software (version 4.1.2) and RStudio (2021.09.0+351 "Ghost Orchid"). Visualisations of distributions of SCr and eGFR results were done using 'ggplot2' package in RStudio (2021.09.0+351 "Ghost Orchid").

## Patient and public involvement

The participants in this study were healthcare clients who visited public health facilities and generated at least one electronic entry in their health record. Retrospective data for this population spanned about 8 years. Inclusion in the study was restricted to healthcare clients who accessed care between 2016 and 2017 and included their complete retrospective data. The

study questions were designed to explore the kidney functions and common comorbidities among these healthcare clients who seek care from public facilities. A waiver for participants' consent was granted by the University of Cape Town Faculty of Health Sciences Ethics Committee because the data were obtained directly from digital routine health data collated by the PHDC and were anonymised and perturbed prior to receipt in order to prevent the possibility of re-identification of participants.

## Results

### Comorbidities in individuals having kidney function tests

An overview of the healthcare clients receiving SCr and eGFR test results is shown by HIV status in Table 1. A total of 45 640 healthcare clients aged 18–80 years were identified as having at least one SCr/eGFR test. Out of these, 22 961 (50.3%) were PLHIV. Healthcare clients with subsequent eGFR results were matched to the SCr test records for further comparison: among the healthcare clients who received SCr laboratory results with or without eGFR results, 32 211 (70.6%) were females and 13 394 (29.3%) were males, the sex status of 35 (0.1%) healthcare clients was not identified. Of the total healthcare clients, 17 729 received only first SCr results without an accompanying eGFR: apart from the high proportions of the HIV-negative healthcare clients and females in this sub-population, there are no particular characteristics indicating why they did not receive eGFR results, and it is likely that this is just a random occurrence in cases where the laboratory service did not receive sufficient additional data to calculate eGFR. The general characteristics, age distribution at first SCr testing, and distribution of first SCr results without matched eGFR results of this sub-population are provided in supplementary data, S1 Table, S1 and S2 Figs.

The median age at first test result for both SCr and eGFR was lower for PLHIV at 33 (IQR: 27–41) years and 36 (IQR: 30–43) years respectively, when compared to HIV-negative

**Table 1. Descriptive statistics for PLHIV and HIV-negative individuals showing median age at first test and the number of individuals with comorbidities (n %) for TB, Hypertension, Diabetes, and CKD.**

| Characteristic | Overall<br>N = 45 640[1] | HIV status<br>HIV-negative<br>N = 22 689[1] | HIV-positive<br>N = 22 951[1] | p-value[2] |
|---|---|---|---|---|
| Median (IQR) age (years) at recruitment | 46 (36–57) | 54 (43–62) | 41 (34–48) | <0.001 |
| Female (n,%) | 32 211 (71) | 15 557 (69) | 16 654 (73) | <0.001 |
| Male (n,%) | 13 394 (29) | 7 111 (31) | 6 283 (27) | |
| Tuberculosis (n,%) | 11 693 (26) | 2 022 (8.9) | 9 671 (42) | <0.001 |
| Hypertension (n,%) | 21 552 (47) | 15 966 (70) | 5 586 (24) | <0.001 |
| Diabetes (n,%) | 9 183 (20) | 7 326 (32) | 1 857 (8.1) | <0.001 |
| CKD (n,%) | 2 394 (5.2) | 1 686 (7.4) | 708 (3.1) | <0.001 |
| Median (IQR) age (years) at first SCr | 39 (30–51) | 49 (37–57) | 33 (27–41) | <0.001 |
| Median (IQR) age (years) at first eGFR | 40 (32–50) | 52 (44–59) | 36 (30–43) | <0.001 |
| Individuals with first SCr that has no eGFR value (n,%) | 17 729 (38.8) | 13 221 (58.3) | 4 508 (19.6) | <0.001 |
| First SCr (μmol/l) | 67 (57–80) | 69 (58–82) | 66 (55–78) | <0.001 |
| First eGFR (mL/min/1.73m$^2$) | 97 (81–117) | 89 (73–107) | 102 (85–122) | <0.001 |
| Deceased (n,%) | 2 327 (5.1) | 1 065 (4.7) | 1 262 (5.5) | <0.001 |
| Median (IQR) age (years) at death | 53 (40–64) | 63 (55–70) | 43 (35–52) | <0.001 |

[1]Median (IQR); n (%)

[2]Wilcoxon rank sum test; Pearson's Chi-squared test.

individuals aged 49 (IQR: 37–57) years and 52 (IQR: 44–59) years, respectively. PLHIV had comparatively lower median SCr results at 66 (IQR: 55–78) µmol/l than HIV-negative individuals at 69 (IQR: 58–82) µmol/l, and this reflected in higher median eGFR results at their first test, accordingly, for PLHIV. Larger proportions of HIV-negative compared to HIV-positive healthcare clients receiving test results presented with hypertension (74.1% vs 25.9%), diabetes (79.8% vs 20.2%), and CKD (70.4% vs 29.6%) (Calculated from Table 1). Those living with HIV were much more likely to have TB (42% vs 8.9%), as expected (Kaplan et al., 2018; Mendelsohn et al., 2022; Swarts et al., 2021). Among the healthcare clients who received SCr laboratory results with or without eGFR results, 2 327 (5.1%) have died. Of 22 951 PLHIV in this dataset 1 262 (5.4%) have died, and of 22 689 HIV-negative healthcare clients, 1 065 (4.7%) have died.

## Kidney function in the healthcare client population

An analysis of the overall age distribution at first SCr and first eGFR test results for females and males per HIV status for all healthcare clients who received test results shows that whilst the age at first SCr and eGFR estimations ranged across 20–65 years for females and males with HIV, those without HIV often had their first test result at much older ages (Fig 1). Though a small portion of HIV-negative males have earlier screening for SCr (Fig 1B), males living with HIV and females in general tend to have SCr tests and eGFR results earlier in life than men without HIV (Fig 1A–1D).

## Stratification of comorbidities and kidney function by age groups

The counts and proportions of common comorbidities per age groups of healthcare clients who received eGFR results were determined. Most common diseases seen in the various age groups were HIV (89%) in the 18–29 years age group, TB (41%) in the 30–44 years age group, hypertension (86%), diabetes (47%) and CKD (24%) in the 60+ years age group (Table 2).

## SCr and eGFR results in healthcare clients by age, sex and HIV status

Analysis of all SCr and eGFR test results for healthcare clients shows how the values change at older ages and differ per HIV status and sex (Fig 2). This comparison shows clearly how the eGFR value adjusts for age and sex so that the difference in SCr results between females and males falls away in corresponding eGFR results (Fig 2A compared to Fig 2B). Similarly, the decrease of kidney function with age is more apparent among PLHIV and HIV-negative individuals when viewing the eGFR result.

## Age stratified analysis of eGFR results in PLHIV and HIV-negative people

The eGFR distribution per age group of healthcare clients who are PLHIV and HIV-negative individuals was analysed. The analysis uses all the eGFR results recorded for all healthcare clients in our study population to make the comparisons and we recognise that multiple results may come from one patient over time. A description of HIV-negative and HIV-positive healthcare clients is shown for each age group. Among the younger and middle age groups, the median eGFR results of the HIV-positive healthcare clients are higher compared to the HIV-negative healthcare clients in these same age groups. At older ages, the difference between the median eGFR results for the HIV-positive and HIV-negative groups is reduced (Fig 3).

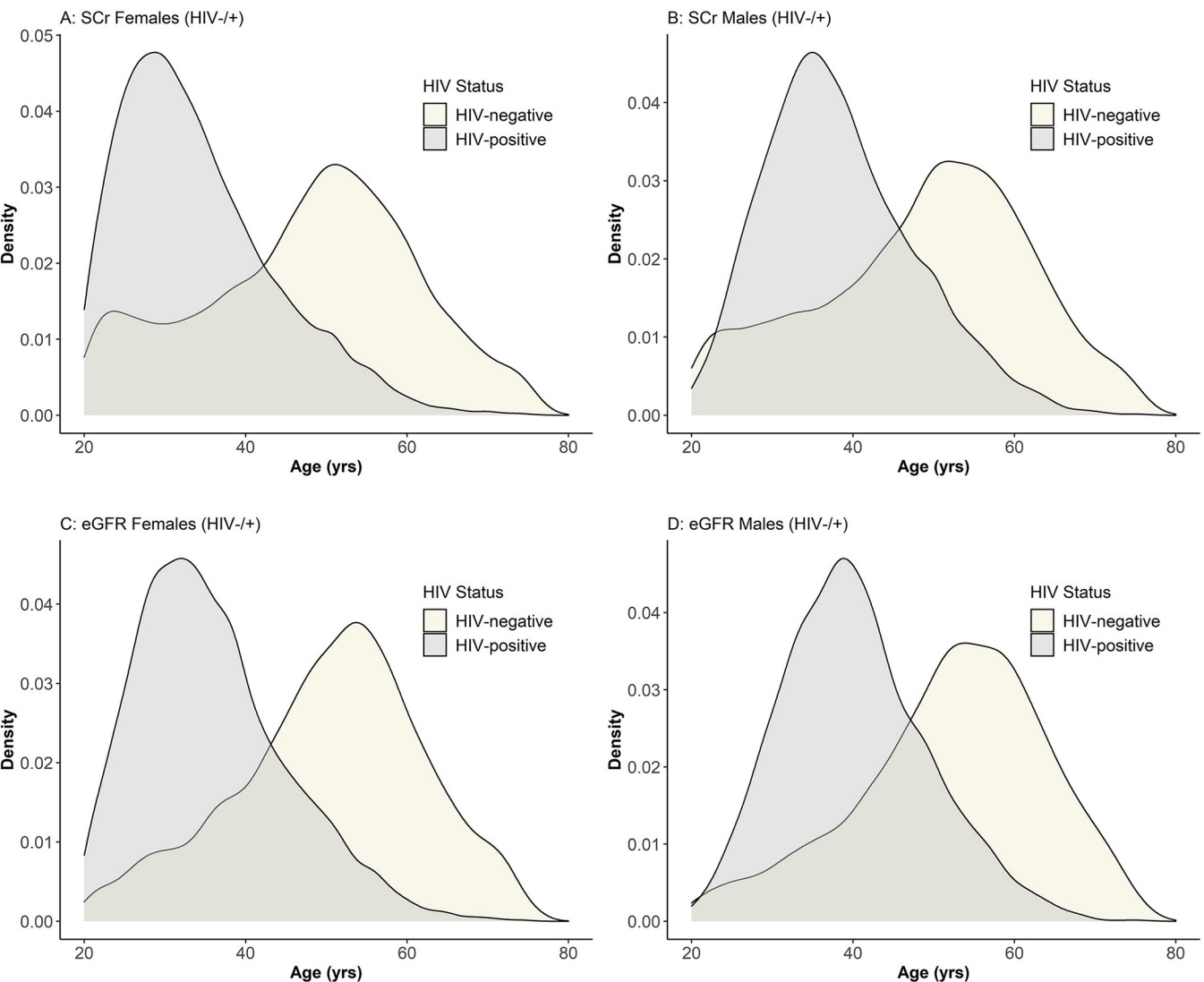

**Fig 1. Distribution of age at first serum creatinine and estimated glomerular filtration rate testing for females and males per HIV status.** The X-axis shows age (years) when the first test results were received, and the Y-axis shows the density distribution. A: Age distribution at first serum creatinine results for females by HIV status. B: Age distribution at first serum creatinine results for males by HIV status C: Age distribution at first estimated glomerular filtration rate results for females by HIV status. D: Age distribution at first estimated glomerular filtration rate results for males by HIV status, showing age (in years) at GFR estimation on the X-axis and the density distribution on Y-axis.

## Age stratified analysis of kidney function with comorbidities TB, hypertension, diabetes, and CKD

The number of all eGFR tests reported for PLHIV and HIV-negative individuals with comorbidities per age categories were analysed. More tests were done for PLHIV than HIV-negative individuals in the 18–29 years category (35 698 vs 11 337); the 30–44 years category (93 505 vs 13 644) and the 45–60 years category (32 353 vs 26 791). Only in the category of 60+ years were more tests reported for HIV-negative individuals (12 763 vs 3 535), but in this category there is a very small number of PLHIV compared to those without HIV (6 916 vs 964).

In each age group the comorbidity with the highest proportion of tests done in that age group reported for healthcare clients with comorbidities were TB (49.2% and 50.6%) for 18–29

**Table 2. Age groups, counts, and proportions of healthcare clients who received eGFR results, showing their ascertained comorbidities.**

| Characteristic | Overall N = 27 907[1] | Age category at first eGFR test | | | | p-value[2] |
| --- | --- | --- | --- | --- | --- | --- |
| | | 18–29 N = 4 968[1] | 30–44 N = 12 276[1] | 45–60 N = 8 456[1] | 60+ N = 2 207[1] | |
| HIV | 18 441 (66) | 4 398 (89) | 10 283 (84) | 3 500 (41) | 260 (12) | <0.001 |
| Tuberculosis | 9 228 (33) | 1 907 (38) | 4 986 (41) | 2 055 (24) | 280 (13) | <0.001 |
| Hypertension | 12 270 (44) | 575 (12) | 3 697 (30) | 6 102 (72) | 1 896 (86) | <0.001 |
| Diabetes | 5 658 (20) | 212 (4.3) | 1 396 (11) | 3 021 (36) | 1 029 (47) | <0.001 |
| CKD | 1 774 (6.4) | 68 (1.4) | 340 (2.8) | 844 (10.0) | 522 (24) | <0.001 |

[1]n (%)

[2]Pearson's Chi-squared test; The percentage in brackets indicates proportion in each age category with the comorbidity of interest; Shaded cells represent the age group in which the comorbidity is most common <0.001.

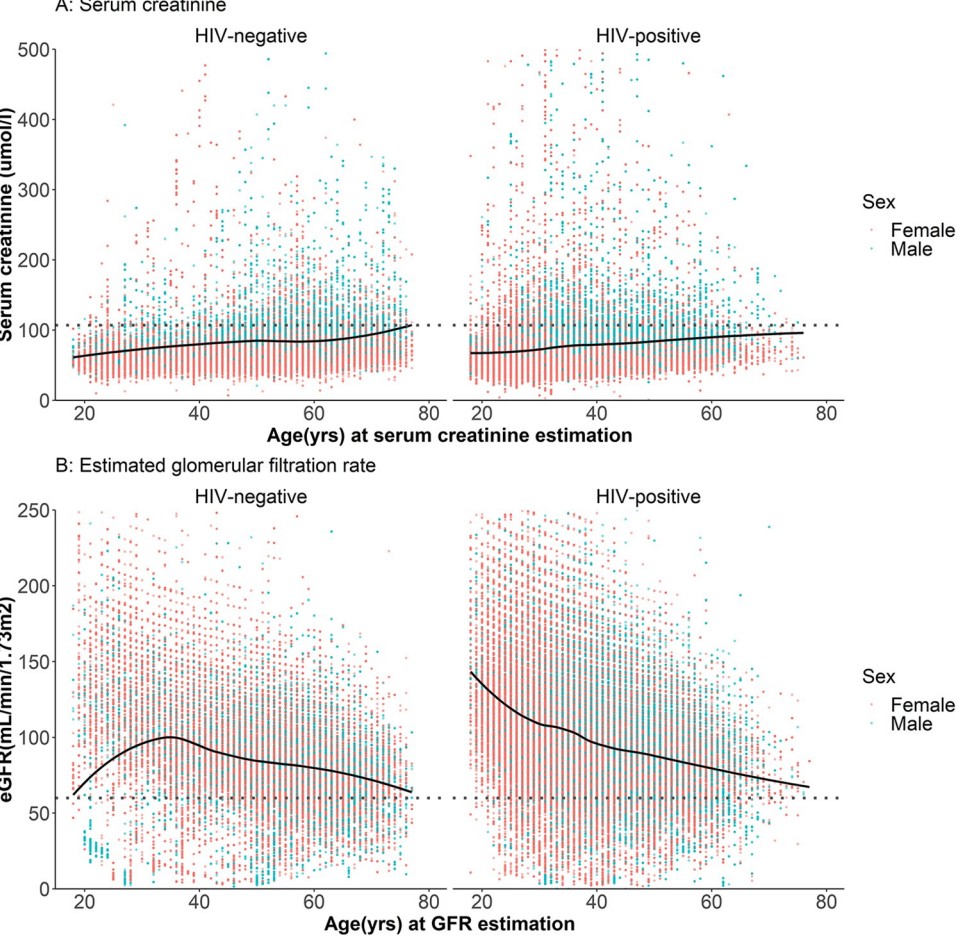

**Fig 2. The distribution of all serum creatinine and estimated glomerular filtration rate results per HIV status and sex.** Data points are coloured by sex. Red: Female, Blue: Male. A. Serum creatinine distribution by HIV status, sex, and age at measurement. Y-axis: Serum creatinine results in μmol/l, X-axis: Age (in years) at creatinine measurement. Dotted line: SCr value above which abnormal kidney function may be inferred. B. Estimated glomerular filtration rate distribution by HIV status, sex, and age (in years) at GFR estimation. Y-axis: eGFR (mL/min/1.73m²), X-axis: Age (years) at eGFR measurement. Dotted line: eGFR value below which an abnormal kidney function is inferred.

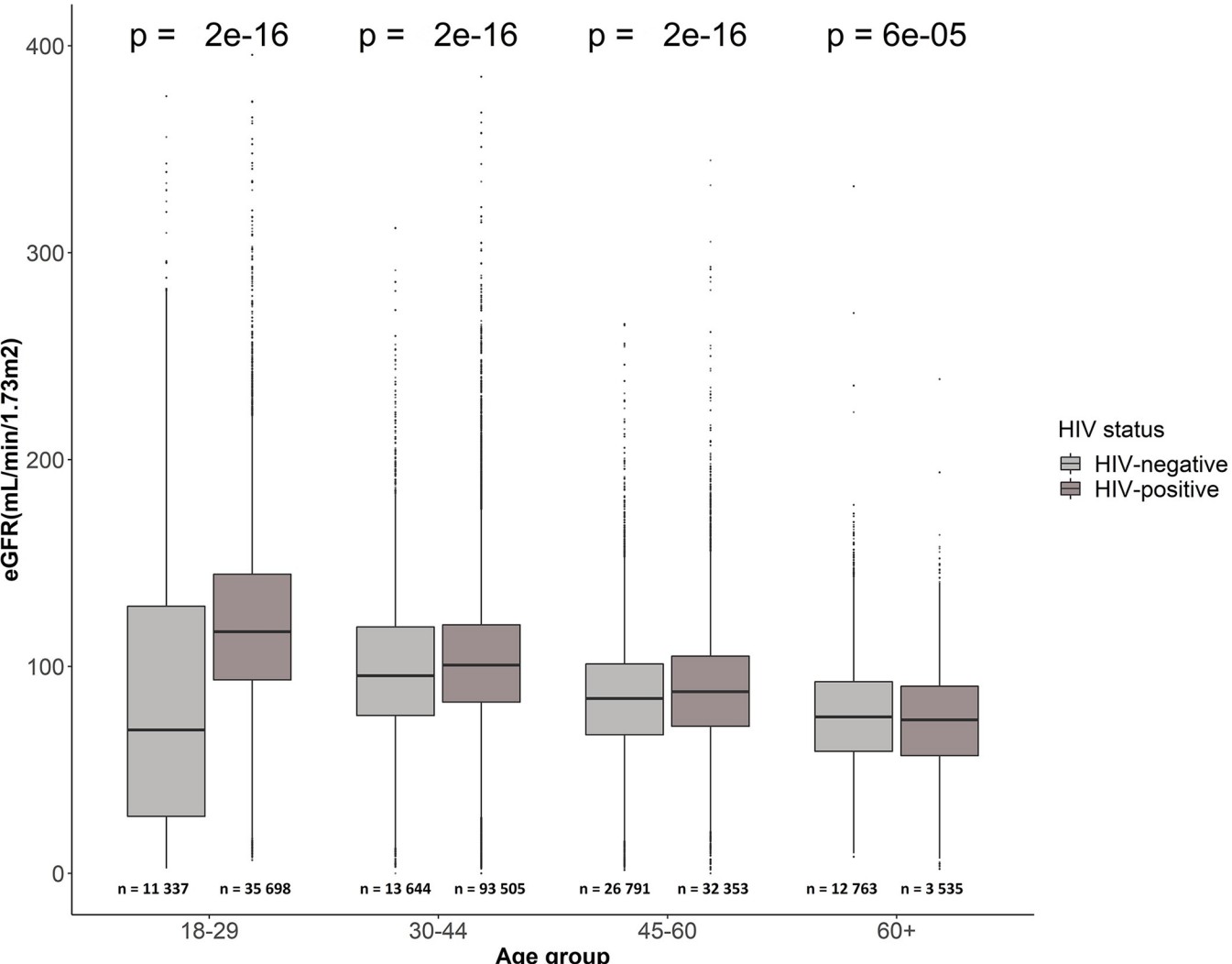

**Fig 3. Distribution of all eGFR results per age groups and HIV status of healthcare clients receiving results.** X-axis: Age groups at eGFR testing; Y-axis: eGFR measures for healthcare clients. Results from PLHIV and HIV-negative healthcare clients in each age group are shown.

years and 30–44 years age groups respectively, and hypertension (67% and 83.4%) for 45–60 years and over 60 years age groups respectively.

An analysis of the median eGFR results for PLHIV and HIV-negative healthcare clients with comorbidities in each age group showed in the younger age categories between 18 and 44 years there is no significant difference in eGFR values in those with and without TB, diabetes and CKD when comparing PLHIV and those who are HIV-negative. Although not statistically significant, PLHIV with hypertension have slightly lower eGFR values than those without HIV (p = 0.06). In the older age categories, however, eGFR is slightly higher for PLHIV who have TB, hypertension and diabetes. For individuals with CKD, the lower eGFR values are similar regardless of HIV status.

## Kidney function trajectories of healthcare clients with HIV and other comorbidities

Whilst previously we analysed all available test results together, we recognise that there are often multiple tests from the same individual, and that the longitudinal trajectories for the

population groups might yield more detailed information about the relationship between patient characteristics and their kidney function test results. The distribution of eGFR results per age groups for individuals who had and those who did not have TB, hypertension and diabetes at the time of eGFR determination, calculating the characteristics, age and comorbidity status of the healthcare client at the time of each test result (Fig 4), provides an overview of these data to gain insight into the eGFR results for healthcare clients who had and did not have the selected comorbidities prior to the linear mixed effect model analysis.

The distribution of eGFR results for healthcare clients who did or did not have TB prior to eGFR testing shows higher median eGFR for clients in the younger age groups (18–29 years and 30–44 years) who had TB and the difference is reduced for healthcare clients in the 45–60 years and over 60 years age groups. For healthcare clients who had hypertension and diabetes, the median eGFR results were lower for those in the 18–29 years age group, with no differences in median eGFR results for those in the 30–44 years, 45–60 years and over 60 years age groups (Fig 4).

The random effect part of the mixed effects model showed considerable variation of eGFR results (kidney function) between healthcare clients, with a standard deviation (SD) of 23.68 mL/min/1.73m$^2$ and residual variance SD of 37.09 mL/min/1.73m$^2$. The fixed effects part of the model is presented in Fig 5.

The fixed effects part of the model (Fig 5) indicates that being in an older age category at eGFR testing is associated with lower eGFR for healthcare clients aged 30–44 years (-10.2 mL/min/1.73m$^2$, 95% Confidence interval [CI]: -11, -9.4), 45–60 years (-23.7 mL/min/1.73m$^2$; 95% CI: -24.7, -22.7), and over 60 years (-33.4 mL/min/1.73m$^2$; 95% CI: -34.9, -31.3) as compared to those aged 18–29 years if sex and comorbidity status remain constant. The model suggests that eGFR is lower for male healthcare clients by an average of -2.8 mL/min/1.73m$^2$ (95% CI: -3.6, -2.0) compared to females. Having hypertension ascertained prior to eGFR testing was found to be associated with lower eGFR results by an average of -1.5 mL/min/1.73m$^2$ (95% CI: -2.1, -0.82) compared to those without hypertension.

Healthcare clients already ascertained with HIV at the time of testing on average have 4.1 mL/min/1.73m$^2$ (95% CI: 3.2, 4.9) higher eGFR results than those who did not have HIV given that age, sex and comorbidity status remain constant. Healthcare clients with diabetes at the time of testing on average have 3.6 mL/min/1.73m$^2$ (95% CI: 2.6, 4.5) higher eGFR results than those without diabetes when age, sex and comorbidity status remain constant. Healthcare clients who have had TB by the time of testing on average have 2.4 mL/min/1.73m$^2$ (95% CI: 1.7, 3.0) higher eGFR results than those who never had TB when age, sex and comorbidity status remain constant.

## Discussion

It can be seen in Fig 1 that HIV-negative and PLHIV sub-populations are quite different in terms of their access to kidney function tests, and are most likely getting these tests for quite different reasons: for PLHIV, it appears that individuals are being tested at a younger age, most likely as part of routine screening which forms part of the standard of care for PLHIV in South Africa; whereas in the HIV-negative population, screening at older ages is much more likely to be in response to presentation of symptoms related to kidney disease. In essence, this suggests that PLHIV are being tested regardless of their apparent health, whereas those without HIV are most likely being tested only if they are symptomatic as a result of other conditions. Earlier female healthcare seeking behaviours compared to later male healthcare seeking behaviours including access to maternal care, high prevalence of CKD among women [27], as well as generally earlier age of HIV infection in women [28], may be the reasons why more females are represented in the study population than males.

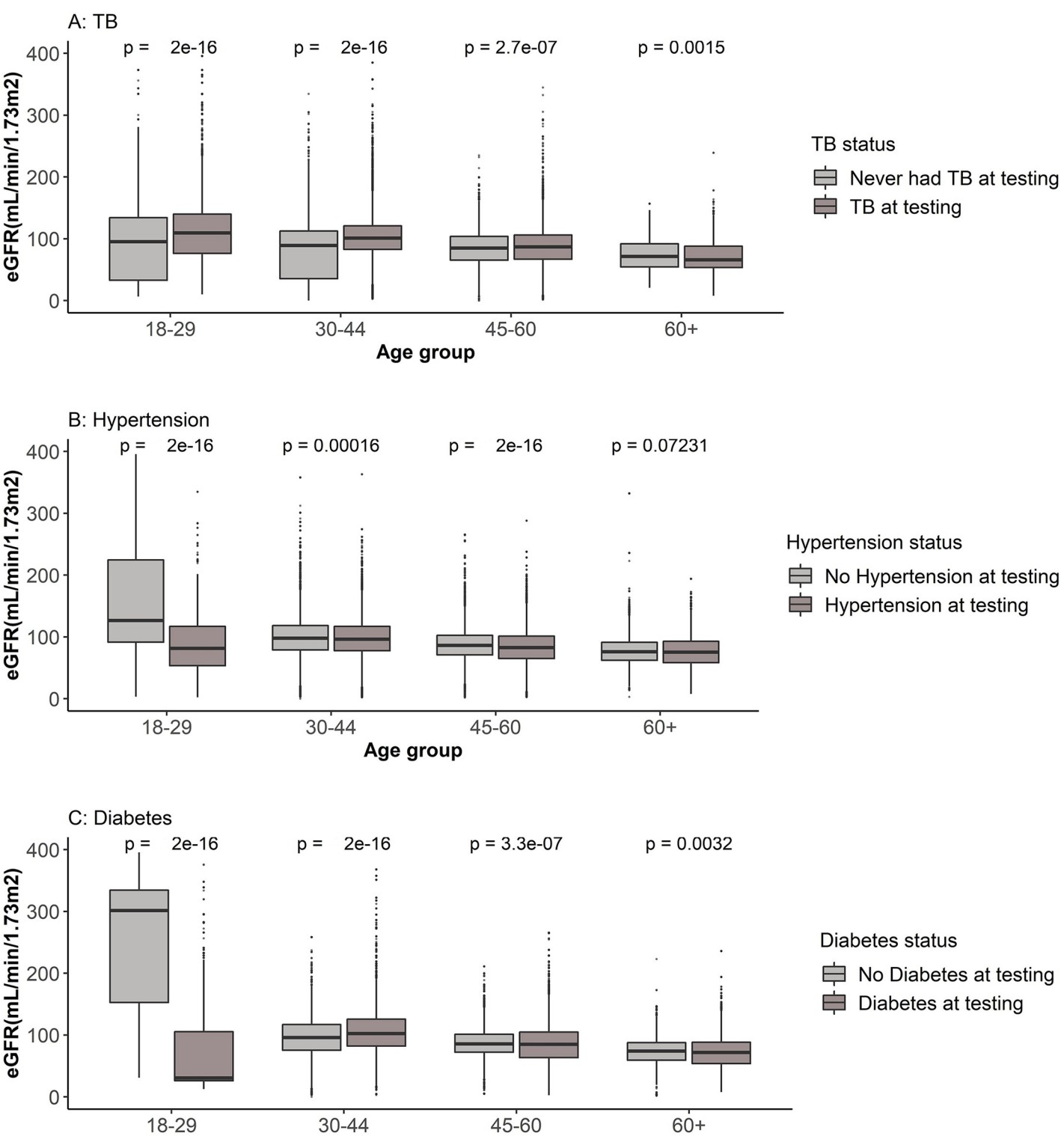

**Fig 4. Distribution of all eGFR results per sex and comorbidity statuses at the time of testing.** X-axis: Age (in groups) at GFR estimation; Y-axis: Estimated glomerular filtration rate in mL/min/1.73m². A. Distribution of eGFR results of healthcare clients who had or did not have TB prior to GFR estimation. B. Distribution of eGFR results of healthcare clients who had or did not have hypertension prior to GFR estimation. C. Distribution of eGFR results of healthcare clients who had or did not have diabetes prior to GFR estimation.

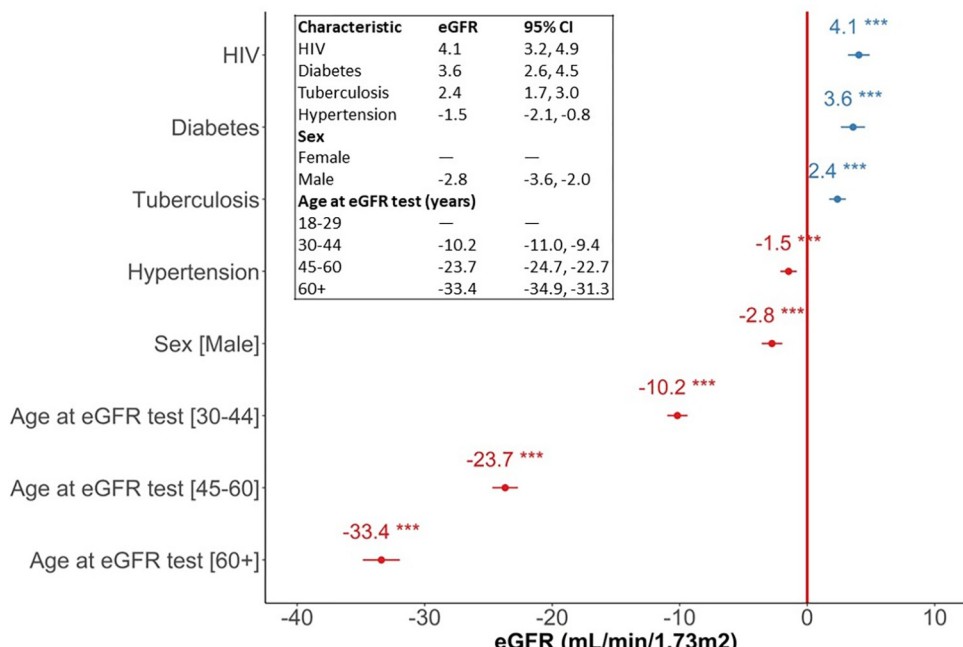

**Fig 5. Linear mixed-effect regression results showing regression coefficients and 95% confidence interval (CI), describing the relationship between eGFR, age category at testing, HIV status and other comorbidities.** X-axis: Regression coefficients (eGFR in mL/min/1.73m$^2$); Y-axis: Predictors.

Early linkage to care and subsequent regular screening for kidney function even without prior related symptoms or illnesses may account for SCr and eGFR results that indicate better kidney function at first testing in PLHIV, whereas HIV-negative individuals may only start getting tested once they are identified as having high risk of renal disease or are already ill and showing symptoms suggestive of renal conditions which more commonly occurs at older ages with the advent of other non-communicable diseases (NCDs). With low median SCr results, median eGFR remains high among young PLHIV even when adjusting for age and sex, which suggests in general there are adequate metabolic processes for the excretion of SCr in these healthcare clients.

Although the overview of all raw SCr results (Fig 2) suggests that HIV-negative younger people having kidney function tests tend to have good kidney function, when adjusted for age and sex to generate eGFR results, it suggests that their kidney function may be more likely to be reduced. This suggests that they may be referred for testing because they are presenting with clinical symptoms related to suspected kidney impairment from other causes. The plotted eGFR results shown in Fig 3 demonstrate this difference in kidney function between the HIV-negative and PLHIV populations in the youngest age brackets.

Poorer kidney function at an older age for first testing for healthcare clients without HIV, reflected in lower eGFR values, may reflect that this population may be presenting at health care facilities with symptoms of chronic conditions which become more common as people age and may impact kidney function. This population may only be receiving kidney function tests because kidney dysfunction is suspected due to their clinical presentation and are therefore presenting with lower kidney function results when first tested.

When PLHIV age, instead of the higher eGFR results seen when they are screened at younger ages, their kidney function results begin to approach those of the HIV-negative groups that have been referred for kidney function tests due to other health conditions, and the eGFR

results between the two groups may not differ as much. This suggests that as this population of healthcare clients with HIV age, their kidney function becomes generally similar to the kidney function of the HIV-negative healthcare clients who have non-HIV-related health conditions that affect their kidney function.

In the younger age brackets not much detectable difference is seen based on comorbidity profiles: PLHIV with the comorbidities analysed in this study do not in general appear to have kidney test results that are significantly different from those in HIV-negative clients of the same age and comorbidity profiles. As already discussed, the HIV-negative group of healthcare clients attending facilities may be presenting with one or more of the comorbidities and are likely to be attending healthcare facilities because they are feeling unwell and have symptoms due to an ongoing condition. This analysis suggests that having HIV combined with these comorbidities is not associated with greater impact on kidney function compared with the impact of these comorbidities on people without HIV.

Considering that the prevalence of HIV and TB is high in South Africa and the burden of these conditions are seen in the young to mid-aged population, PLHIV and individuals presenting with TB who have kidney function tests would most likely be in the young and middle age groups, as seen in our study population (Table 2). Given the high prevalence of hypertension [29] and diabetes [30] in South Africa, it is expected that higher proportions of HIV-negative healthcare clients receiving kidney function tests but not being tested as part of routine HIV/TB care may generally reflect in the mid- to older -age population in which other chronic NCDs may be determining their healthcare seeking behaviours. Ageing with hypertension, diabetes, and CKD are known to be significant risk factors for renal disorders and this population are most likely having eGFR tests due to the impacts of these common comorbidities on kidney function. The HIV-negative subgroup of healthcare clients in this population is therefore not representative of the general HIV-negative population, as it is strongly biased towards those with other health conditions.

In the older age groups, there is not a substantial difference in average eGFR values for PLHIV and HIV-negative individuals with similar comorbidity profiles. Whilst we may have anticipated worse eGFR results in PLHIV [31], our observation in this study could be due to the much smaller number of individuals in the older age group of PLHIV–many more people with severe HIV-related disease may not have lived to this age group given the poor availability of ART when they were younger. This is supported by the different median age at death between PLHIV, at 43 (IQR: 35–52) years and HIV-negative individuals at 63 (IQR: 55–70) years (Table 1). In addition, the proportion of eGFR tests for PLHIV in each age bracket (Table 3) are much lower in older age brackets when compared to number of tests in people without HIV who are attending healthcare facilities with other conditions. For example, in the oldest age bracket, there are only approximately 3 500 tests from PLHIV compared to 12 763 tests in the HIV-negative group in this age bracket.

Those PLHIV who have survived to the older age brackets have been screened, have perhaps had better access to health care and ART than others diagnosed at the same time, and may have experienced more mild disease or delayed onset of HIV-related symptoms. In addition, better linkage to care and/or healthcare seeking behaviours may also predispose them to better management of their additional chronic conditions and comorbidities alongside managing their HIV infection [32].

As expected, our analysis also shows that receiving treatment for hypertension is associated with lower eGFR results. In this study, those ascertained with hypertension are identified in the PHDC data because they are receiving anti-hypertensive medications, whereas we have no way to identify unmedicated hypertensive individuals in the absence of clinical observational data. Previous epidemiological studies have reported that risks of cardiovascular diseases and

**Table 3. Median (IQR) eGFR for all tests done per age group and HIV status of healthcare clients with comorbidities.**

| Age group: 18–29 years | | Median (IQR) eGFR value[1] | | | |
|---|---|---|---|---|---|
| Comorbidity | Counts (% of all tests) [1] | Overall | HIV-negative | HIV-positive | p-value[2] |
| Tuberculosis | 23 124 (49.2) | 118 (99–141) | 118 (99–146) | 118 (97–141) | 0.90 |
| Hypertension | 11 438 (24.3) | 117 (96–146) | 123 (100–153) | 116 (95–143) | 0.06 |
| Diabetes | 10 164 (21.6) | 128 (107–153) | 136 (103–163) | 125 (107–142) | 0.15 |
| CKD | 9 865 (21) | 75 (73–99) | 73 (39–85) | 76 (45–107) | 0.30 |
| **Age group: 30–44 years** | | | | | |
| Comorbidity | Counts (% of all tests) [1] | Overall | HIV-negative | HIV-positive | p-value[2] |
| Tuberculosis | 54 171 (50.6) | 105 (86–124) | 106 (88–126) | 105 (89–124) | 0.60 |
| Hypertension | 33 078 (30.9) | 101 (85–121) | 100 (85–125) | 102 (85–120) | 0.90 |
| Diabetes | 13 122 (12.2) | 106 (89–128) | 105 (89–130) | 107 (89–126) | 0.70 |
| CKD | 10 545 (9.8) | 59 (43–79) | 57 (40–79) | 61 (44–79) | 0.20 |
| **Age group: 45–60 years** | | | | | |
| Comorbidity | Counts (% of all tests) [1] | Overall | HIV-negative | HIV-positive | p-value[2] |
| Tuberculosis | 20 514 (34) | 94 (77–112) | 89 (70–109) | 95 (79–112) | <0.001 |
| Hypertension | 39 620 (67) | 88 (74–103) | 87 (73–102) | 89 (75–106) | <0.001 |
| Diabetes | 20 869 (35.3) | 90 (74–107) | 89 (74–106) | 91 (75–112) | 0.026 |
| CKD | 11 234 (19) | 59 (46–72) | 58 (45–72) | 60 (48–72) | 0.30 |
| **Age group: 60+ years** | | | | | |
| Comorbidity | Counts (% of all tests) [1] | Overall | HIV-negative | HIV-positive | p-value[2] |
| Tuberculosis | 2 774 (17) | 83 (66–99) | 76 (62–94) | 87 (74–102) | <0.001 |
| Hypertension | 13 585 (83.4) | 77 (63–92) | 76 (63–91) | 84 (67–99) | <0.001 |
| Diabetes | 7 834 (48.1) | 77 (60–93) | 76 (60–92) | 83 (64–100) | 0.035 |
| CKD | 5 802 (35.6) | 55 (44–66) | 55 (44–66) | 57 (46–66) | 0.40 |

[1]Median (IQR); n (%);

[2]Wilcoxon rank sum test; Shaded cells represent statistically significant difference between PLHIV and HIV-negative individuals with the comorbidity.

events may increase in individuals with renal impairments [33,34], but we are unable to evaluate the full impact of hypertension on this population because of our inability to identify people with hypertension who are not receiving medications from the available data.

We did not include CKD in the mixed effects model because the ascertainment of CKD by the PHDC is largely dependent on longitudinal analysis of SCr/eGFR laboratory test results and may have caused modelling problems related to multicollinearity. We also note that the algorithm does not currently identify individuals with persistent proteinuria with preserved eGFR who may still qualify as having CKD. The results from the mixed effects model suggest that older age categories are associated with substantially lower eGFR results, as expected. Being male, and having hypertension were also associated with slightly lower eGFR values. In this population of healthcare clients, having HIV, TB and diabetes are associated with slightly higher eGFR results. People with TB are often also living with HIV, and therefore may also be subject to increased screening prior to having any symptoms of renal disease, and people living with diabetes are also linked to care and undergo screening for kidney function. Our observations therefore may reflect that being linked to care and having screening for kidney function for patients with these comorbidities may result in more frequent kidney function tests before kidney function has deteriorated, with corresponding higher eGFR test results.

A major limitation in this study is bias in the whole study population which is enriched for PLHIV and HIV-negative individuals who may be accessing care for various reasons such as accessing maternal care or accessing care because of ill health due to TB or chronic NCDs. The routine health data used in this study had a limited number of disease conditions ascertained by the PHDC and did not include important comorbidities like cardiovascular diseases (CVD). There are also challenges related to data linkage due to poor record keeping at facilities, which might result in situations where records cannot be linked or improperly linked together [35,36].

## Conclusion

Many clinical studies have shown the relationship between HIV and subsequent nephropathy [10,37], as well as reporting improvements in kidney function for PLHIV after ART initiation [11,12]. In this study we have assessed kidney function testing and results in a population of healthcare clients in the Western Cape, South Africa, and as such are reporting on testing practices in the healthcare client population, rather than the aetiological processes by which HIV can lead to the development of other comorbidities.

In summary, in this study the ageing population of PLHIV who are actively seeking healthcare appear quite similar in comorbidities and kidney function to those without HIV, and it is likely that this is due to the history of access to ART in South Africa when these older individuals were first diagnosed with HIV. Whilst the older HIV population currently looks similar to the older HIV-negative population in terms of kidney disease and associated healthcare requirements, we still cannot accurately predict the future health requirements of those who linked to ART whilst young, as they age.

Our analysis of kidney test results in younger healthcare clients suggests routine kidney function screening is being undertaken for young PLHIV in line with current guidelines and best practice, but in future studies we also need to identify where screening is not reaching those who might benefit from it.

The value of this study is that we have used real data about routine kidney health testing from healthcare clients in Khayelitsha, which offer insights into what is actually happening in facilities with respect to kidney function testing. This is useful for evaluating the implementation of treatment guidelines and healthcare delivery policies and service utilisation for PLHIV.

## Supporting information

**S1 Fig. Distribution of age at first serum creatinine testing without matched estimated glomerular filtration rate for females and males per HIV status.** X-axis: Age (years) of first test results. Y-axis: Density distribution. A: Age distribution at first serum creatinine results for females by HIV status. B: Age distribution at first serum creatinine results for males by HIV status.
(PDF)

**S2 Fig. Distribution of first serum creatinine without matched estimated glomerular filtration rate results per HIV status and sex.** Data points are coloured by sex. Red: Female, Blue: Male. X-axis: Age (in years) at creatinine estimation. Y-axis: Serum creatinine results (umol/l). The dotted line shows the SCr value above which abnormal kidney function may be inferred.
(PDF)

**S1 Table. Characteristics of healthcare clients who received only SCr results without eGFR.**
(PDF)

## Acknowledgments

We acknowledge the Provincial Health Data Centre, Health Impact Assessment Directorate of the Western Cape Government Health Department for the provision of the anonymized Khayelitsha dataset.

## Author Contributions

**Conceptualization:** Nicki Tiffin.

**Data curation:** Richard Osei-Yeboah.

**Formal analysis:** Richard Osei-Yeboah, Olina Ngwenya.

**Funding acquisition:** Nicki Tiffin.

**Investigation:** Richard Osei-Yeboah, Nicki Tiffin.

**Methodology:** Richard Osei-Yeboah, Olina Ngwenya, Nicki Tiffin.

**Resources:** Nicki Tiffin.

**Supervision:** Nicki Tiffin.

**Validation:** Richard Osei-Yeboah, Olina Ngwenya, Nicki Tiffin.

**Visualization:** Richard Osei-Yeboah.

**Writing – original draft:** Richard Osei-Yeboah.

**Writing – review & editing:** Richard Osei-Yeboah, Olina Ngwenya, Nicki Tiffin.

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
