## [Decision Letter · Decision Letter 0]

22 Nov 2023

PGPH-D-23-01846

Kidney function in healthcare clients in Khayelitsha, South Africa: Routine laboratory testing and results reflect distinct healthcare experiences by age for healthcare clients with and without HIV.

Dear Dr. Tiffin,

Thank you for submitting your manuscript to PLOS Global Public Health. After careful consideration, we feel that it has merit but does not fully meet PLOS Global Public Health’s publication criteria as it currently stands. Therefore, we invite you to submit a revised version of the manuscript that addresses the points raised during the review process.

The manuscript has been evaluated by two reviewers, and their comments are available below.

The reviewers have raised a number of minor concerns. Specifically, they feel that the discussion and conclusion could be condensed for clarity, and that a discussion of study limitations would increase the quality of the manuscript.

Could you please carefully revise the manuscript to address all comments raised?

I have also noted that one or more reviewers has recommended that you cite specific previously published works. As always, we recommend that you please review and evaluate the requested works to determine whether they are relevant and should be cited. It is not a requirement to cite these works.

We look forward to receiving your revised manuscript.

Kind regards,

Johanna Pruller, Ph.D.

PLOS Staff Editor

Journal Requirements:

2. We have noticed that you have uploaded Supporting Information files, but you have not included a list of legends. Please add a full list of legends for your Supporting Information files after the references list.

Additional Editor Comments (if provided):

Reviewers' comments:

Reviewer's Responses to Questions

**Comments to the Author**

1. Does this manuscript meet PLOS Global Public Health’s publication criteria? Is the manuscript technically sound, and do the data support the conclusions? The manuscript must describe methodologically and ethically rigorous research with conclusions that are appropriately drawn based on the data presented.

Reviewer #1: Yes

Reviewer #2: Yes

2. Has the statistical analysis been performed appropriately and rigorously?

Reviewer #1: Yes

Reviewer #2: Yes

3. Have the authors made all data underlying the findings in their manuscript fully available (please refer to the Data Availability Statement at the start of the manuscript PDF file)?

Reviewer #1: Yes

Reviewer #2: No

4. Is the manuscript presented in an intelligible fashion and written in standard English?

Reviewer #1: Yes

Reviewer #2: Yes

5. Review Comments to the Author

Reviewer #1: This study investigated serum creatinine and estimated glomerular filtration rate (eGFR) results in individuals living with HIV (PLHIV) and the HIV-negative population accessing healthcare facilities in Khayelitsha, South Africa during 2016/2017. The analysis included stratification by sex, age, and the presence of comorbidities such as tuberculosis, hypertension, chronic kidney disease (CKD), and diabetes. The study's significance lies in its utilization of authentic data from routine kidney health testing among healthcare clients in Khayelitsha, providing valuable insights into the actual practices within facilities regarding kidney function testing.

The introduction of the study is appropriate, featuring a suitable presentation of the study's background and well-explained objectives.

The study methodology is deemed appropriate and repeatable.

The results are well presented. One minor comment on the section "Age stratified analysis of kidney function with comorbidities TB, hypertension, diabetes, and CKD": the sentence "More tests were done for PLHIV than HIV-negative people in the 18-29 years category (35,698 vs 11,337); the 30-44 years category (93,505 vs 13,644); and the 45-60 years category (32,353 vs 26,791)" appears to incorrectly align the numbers within the brackets with the corresponding categories.

Regarding the presentation of p-values in Figure 3 and Figure 4, it might be clearer to represent them as “p =” rather than “p =<”.

The discussion is based on the results obtained from the study, providing appropriate analysis and comparison with similar studies. It would be valuable to discuss briefly the limitations of the study.

While the conclusion provides a thorough description, it could benefit from being condensed for brevity and clarity. Additionally, ensuring consistency between the conclusion at the end of the article and the conclusion in the abstract would contribute to a cohesive presentation of the study.

Thank you for considering these comments.

Reviewer #2: Kidney function in healthcare clients in Khayelitsha, South Africa: Routine laboratory testing and results reflect distinct healthcare experiences by age for healthcare clients with and without HIV

Summary

This is a study assessing SCr and estimated glomerular filtration rate (eGFR) results for PLHIV and HIV-negative healthcare clients aged 18-80 years accessing healthcare in Khayelitsha, South Africa. The authors explore the relationship between these renal function tests and several other patient characteristics including age, infectious diseases (TB and HIV status), and non-communicable diseases more common in older individuals. The results show that people living with HIV (PLWH) tended to have better renal function, particularly among the younger age groups, but this difference attenuated with increasing age. The main reason for the differences is the indication for testing with renal function testing being a routine requirement for otherwise healthy PLWH, but among the HIV negative population likely necessitated by illness that might affect renal function.

General comment

This is a well written manuscript, and the authors must be commended for putting together such a thorough manuscript!

Specific comments

PHDC disease episode definition:

“Microbiology culture-Toxocara canis” – This seems an unlikely target for the diagnosis of tuberculosis. Did the authors mean Micobacterium tuberculosis?

“High confidence definition of hypertension includes dispensed hydrochlorothiazide.” – Is this the only antihypertensive in use in South Africa or perhaps the most common? Would ascertainment have been improved by adding other commonly used antihypertensives?

Data analysis

“The counts and proportions of the healthcare clients receiving eGFR results, and the ascertained comorbidities per 18-29 years, 30-44 years, 45-60 years, and over 60 years age groups were described” – Is there any reason for the choice of these specific age bands?

Discussion

This manuscript could have benefited from a paragraph dedicated to the strengths and weaknesses of this study. It is unclear to me whether all the PLWH were tested at ART initiation or at random points during their treatment course. It also seems the authors assumed the measurements were at random points. To address this question, it might be helpful to add to the methods section the monitoring algorithm according to the national ART guidelines for PLWH. Currently, the first line in the discussion provides a new guideline, but this new information is best placed in the methods section. It would also be helpful to discuss the question of incident HIV among older people in South Africa, and perhaps the challenge of delayed presentation among these older people (https://pubmed.ncbi.nlm.nih.gov/35218732/).

Although there might be no word limit, the conclusion is quite long and the discussion can be a bit more concise.

6. PLOS authors have the option to publish the peer review history of their article (what does this mean?). If published, this will include your full peer review and any attached files.

**Do you want your identity to be public for this peer review?** For information about this choice, including consent withdrawal, please see our Privacy Policy.

Reviewer #1: No

Reviewer #2: No

---

## [Decision Letter · Decision Letter 1]

27 Feb 2024

PGPH-D-23-01846R1

Kidney function in healthcare clients in Khayelitsha, South Africa: Routine laboratory testing and results reflect distinct healthcare experiences by age for healthcare clients with and without HIV.

Dear Dr. Tiffin,

Thank you for submitting your manuscript to PLOS Global Public Health. After careful consideration, we feel that it has merit but does not fully meet PLOS Global Public Health’s publication criteria as it currently stands. Therefore, we invite you to submit a revised version of the manuscript that addresses the points raised during the review process.

We look forward to receiving your revised manuscript.

Kind regards,

Miquel Vall-llosera Camps

Staff Editor

Journal Requirements:

Reviewers' comments:

Reviewer's Responses to Questions

**Comments to the Author**

1. If the authors have adequately addressed your comments raised in a previous round of review and you feel that this manuscript is now acceptable for publication, you may indicate that here to bypass the “Comments to the Author” section, enter your conflict of interest statement in the “Confidential to Editor” section, and submit your "Accept" recommendation.

Reviewer #1: (No Response)

2. Does this manuscript meet PLOS Global Public Health’s publication criteria? Is the manuscript technically sound, and do the data support the conclusions? The manuscript must describe methodologically and ethically rigorous research with conclusions that are appropriately drawn based on the data presented.

Reviewer #1: Yes

3. Has the statistical analysis been performed appropriately and rigorously?

Reviewer #1: Yes

4. Have the authors made all data underlying the findings in their manuscript fully available (please refer to the Data Availability Statement at the start of the manuscript PDF file)?

Reviewer #1: Yes

5. Is the manuscript presented in an intelligible fashion and written in standard English?

Reviewer #1: Yes

6. Review Comments to the Author

Reviewer #1: Thank you for the thorough review and for addressing my comments. However, there is still some divergence between the conclusion at the end of the article and that presented in the abstract. I kindly request a closer alignment between the conclusions in the article and the abstract for consistency.

7. PLOS authors have the option to publish the peer review history of their article (what does this mean?). If published, this will include your full peer review and any attached files.

**Do you want your identity to be public for this peer review?** For information about this choice, including consent withdrawal, please see our Privacy Policy.

Reviewer #1: No

---

## [Editor Report · Decision Letter 2]

12 Apr 2024

Kidney function in healthcare clients in Khayelitsha, South Africa: Routine laboratory testing and results reflect distinct healthcare experiences by age for healthcare clients with and without HIV.

PGPH-D-23-01846R2

Dear Professor Tiffin,

We are pleased to inform you that your manuscript 'Kidney function in healthcare clients in Khayelitsha, South Africa: Routine laboratory testing and results reflect distinct healthcare experiences by age for healthcare clients with and without HIV.' has been provisionally accepted for publication in PLOS Global Public Health.

Best regards,

Julia Robinson

Executive Editor